chemical biology

lignin, dehydrogenative polymerization, dilignol, β-5 structure, peroxidase

**Author for correspondence:**
Yasuyuki Matsushita
e-mail: ysmatsu@agr.nagoya-u.ac.jp

This article has been edited by the Royal Society of Chemistry, including the commissioning, peer review process and editorial aspects up to the point of acceptance.

# Unexpected polymerization mechanism of dilignol in the lignin growing

Yasuyuki Matsushita, Yuto Oyabu, Dan Aoki
and Kazuhiko Fukushima

Graduate School of Bioagricultural Sciences, Nagoya University, Furo-cho, Chikusa-ku, Nagoya, Aichi 464-8601, Japan

 YM, 0000-0003-1357-3927

Lignin is an essential component of higher plants, which is built by the enzymatic dehydrogenative polymerization of monolignols. First, monolignol is enzymatically oxidized to produce the phenoxy radical, which can form resonance hybrids. Two radical resonant hybrids are coupled with each other to yield dilignol with various linkage types, of which the main structures are β-O-4′ (**I**), β-5′ (**II**) and β-β′ (**III**). However, the reaction mechanism behind the addition lignol radicals to dilignol is not yet fully understood. Here, we show an unexpected reaction with structure **II** during enzymatic dehydrogenative polymerization, which involves cleavage of a covalent linkage and creation of a new radical coupling site. This implied that the β-5 dilignol diversifies the growing pattern of lignin. This discovery elucidates a novel mechanism in lignin polymerization.

## 1. Introduction

Lignocellulosic biomass exhibits a carbon neutral characteristic and thus should be used effectively to reduce carbon dioxide emission, which is one of the causes of global warming. Lignocellulosic biomass is composed of polysaccharides, such as cellulose, and a phenolic polymer, lignin. While cellulose is used in the production of fibres and paper, the use of lignin is limited. One of the reasons is the complexity of its chemical structure. It is well known that lignin is a phenolic natural polymer and does not contain a definite repeating unit [1,2]. Therefore, understanding the structure of lignin is essential for its effective use. In this study, we carry out lignin biosynthesis to help understand its structure.

The biosynthesis and the structure of lignin have yet to be elucidated and have attracted the interest of many research laboratories. The current prevailing theory is that lignin is formed by a repeating radical coupling reaction of lignols by enzymatic dehydrogenative radicalization. The presence of various

**Figure 1.** Enzymatic dehydrogenative polymerization of monolignol. (*a*) Dimerization of coniferyl alcohol. Three major dimers are generated. (*b*) Radicalization of the dimers. **II** is thought to be cleavage of α-*O*-4′ linkage, which is an unknown reaction mechanism.

monolignol species and the frequency of various linkage types are what characterize the resultant lignin macromolecule; however, the details on how the biosynthesis is orchestrated in plants are unclear.

During the initial steps of lignin formation, there are three main types of dilignols, namely, β-*O*-4′ (**I**), β-5′ (**II**) and β-β′ (**III**), that are generated by the coupling of two coniferyl alcohols (figure 1*a*). Next, the dilignol undergoes radicalization by enzymatic oxidation. Once the dilignol radical resonant hybrids form, the radical localizes at phenolic oxygen (C4-O) or ring 5 position (C5), as shown in figure 1*b* [3–6]. However, in the case of **II**, there is the possibility of cleavage at the α-*O*-4′ linkage during the reaction, leading to the generation of new phenoxy (C4′-O) radical. Furthermore, if this hypothesis holds true, the radical can localize at the β′ position with the β′ carbon participating in the lignin polymerization reaction. To our knowledge, this is a novel hypothesis that has not been reported.

**Figure 2.** Synthesis of 13C labelled β-5 dilignol $^{13}$**C-II**. Regents and conditions, (*a*) malonic acid, pyridine, 60°C, 24 h, 72%, (*b*) TMSCl, MeOH, reflux, 1 h, 96%, (*c*) Ag$_2$O, DCM, r.t., 24 h, 36%, (*d*) Ac$_2$O, pyridine, r.t., 24 h, 98%, (*e*) RuCl$_3$, NaIO$_4$, EtOAc/MeCN/H$_2$O, 0°C, 3 h, 58%, (*f*) [2-$^{13}$C]malonic acid, pyridine, 60°C, 24 h, (*g*) TMSCl, MeOH, reflux, 1 h, 19% from **7**, (*h*) LiAlH$_4$, THF, r.t., 1 h, 41%.

To investigate whether this hypothetical reaction occurs during lignin growth, we synthesized **II** and **II** labelled with $^{13}$C at the β′-position ($^{13}$**C-II**), and subjected them to enzymatic dehydrogenative polymerization using the horseradish peroxidase-H$_2$O$_2$ system, which is commonly used for *in vitro* synthesis of artificial lignin for structural analysis [7,8]. The reaction products, i.e. dehydrogenative polymers of **II** (**DHP-II**) and (**DHP-$^{13}$C-II**) were analysed by NMR measurement.

# 2. Material and methods

## 2.1. Synthesis of **II** and $^{13}$**C-II**

**II** and $^{13}$**C-II** were prepared according to the previous reports [9–11]. A schematic of dehydrodiconiferyl alcohol synthesis is shown in figure 2.

Vanillin (**1**) (5 g) was condensed with malonic acid (4.3 g), pyridine (5 ml) and a small amount of piperidine (10 drops) at 60°C for 24 h. After acidification using hydrochloric acid, the reaction mixture was extracted with ethyl acetate. The organic layer was extracted with 20% sodium hydrogen sulfite to remove unreacted vanillin, washed with brine and dried with anhydrous sodium sulfate. The organic solvent was removed at reduced pressure to obtain ferulic acid (**2**) (yield 72%). $^{1}$H NMR (in Acetone-d6) *δ*: 3.93 (3H, *s*), 6.40 (1H, *d*, *J* = 15.9 Hz), 6.88 (1H, *d*, *J* = 8.4 Hz), 7.16 (1H, *dd*, *J* = 8.4 Hz, 2.0 Hz), 7.34 (1H, *d*, *J* = 1.6 Hz), 7.62 (1H, *d*, *J* = 15.9 Hz) ; $^{13}$C-NMR *δ*: 56.3, 111.3, 116.0, 116.1, 123.9, 127.5, 145.9, 148.8, 150.0, 168.3.

Dry methanol (60 ml) was cooled to 0°C under argon gas. Trimethylsilyl chloride (2.9 ml) was added to the solution and stirred for 20 min. Prepared ferulic acid (**2**) (3 g) was added to the methanol solution and heated at reflux (90°C) for 1 h. After cooling to room temperature, the reaction mixture was dissolved in dichloromethane and washed with distilled water and brine prior to being dried over anhydrous sodium sulfate. The organic solvent was evaporated at reduced pressure to obtain methyl ferulate (**3**) (yield 96%). [1]H NMR (in Acetone-d6) $\delta$: 3.72 (3H, s), 3.92 (3H, s), 6.40 (1H, d, J = 16.0 Hz), 6.88 (1H, d, J = 8.4 Hz), 7.15 (1H, dd, J = 8.4 Hz, 2.0 Hz), 7.34 (1H, d, J = 1.6 Hz), 7.60 (1H, d, J = 16.0 Hz) ; [13]C-NMR $\delta$: 51.5, 56.3, 111.3, 115.5, 116.1, 123.9, 127.4, 145.7, 148.7, 150.1, 167.9.

Methyl felulate was dissolved in dichloromethane under argon gas. Finely ground silver oxide (I) (560 mg) was added while stirring at room temperature for 24 h in the dark. The inorganic materials were filtered with celite and rinsed with dichloromethane and hot acetone. The filtrate was then concentrated under reduced pressure and purified by silica gel column chromatography using a hexane-ethyl acetate mixture as an elution solvent to obtain compound **4** (yield 36%). [1]H NMR (in Acetone-d6) $\delta$: 3.73 (3H, s), 3.81 (3H, s), 3.84 (3H, s), 3.92 (3H, s), 4.47 (1H, d, J = 7.9 Hz), 6.04 (1H, d, J = 7.9 Hz), 6.44 (1H, d, J = 16.0 Hz), 6.84 (1H, d, J = 8.1 Hz), 6.91 (1H, dd, J = 8.2 Hz, 2.0 Hz), 7.10 (1H, d, J = 1.9 Hz), 7.29 (1H, s), 7.33 (1H, s), 7.63 (1H, d, J = 16.0 Hz); [13]C-NMR $\delta$: 51.6, 53.0, 55.9, 56.3, 56.5, 88.4, 110.8 113.5, 115.8, 116.3, 119.0, 120.2, 127.4, 129.4, 132.0, 145.4, 145.8, 148.0, 148.6, 151.0, 167.8, 171.7.

Compound **4** (0.57 g) was dissolved in anhydrous pyridine (2.9 ml) and mixed with acetic anhydride (1.4 ml) while stirring at room temperature for 24 h. A small amount of ice was added to the reaction and the reaction solution was extracted with ethyl acetate. The reaction mixture was subsequently washed with acidified water, distilled water, basic water, distilled water and brine, and dried with anhydrous sodium sulfate. The solvent was removed at reduced pressure to obtain compound **5** (yield 98%). [1]H NMR (in Acetone-d6) $\delta$: 2.24 (3H, s), 3.73 (3H, s), 3.82 (3H, s), 3.82 (3H, s), 3.94 (3H, s), 4.50 (1H, d, J = 7.6 Hz), 6.14 (1H, d, J = 7.6 Hz), 7.05 (1H, dd, J = 8.1 Hz, 1.8 Hz), 7.08 (1H, d, J = 8.1 Hz), 7.24 (1H, d, J = 2.0 Hz), 7.31 (1H, s), 7.35 (1H, s), 7.45 (1H, d, J = 16.0 Hz), 7.63 (1H, d, J = 16.0 Hz); [13]C-NMR $\delta$: 20.4, 51.6, 53.0, 56.0, 56.3, 56.5, 87.5, 111.3, 116.4, 119.0, 119.0, 123.9, 124.6, 127.0, 129.7, 139.6, 141.1, 145.3, 145.8, 150.4, 167.7, 171.5, 172.1.

Sodium periodate (0.15 g), water (188 µl) and sulfuric acid (93 µl) were combined in a round-bottom flask and cooled to 0°C while stirring. Ruthenium (III) chloride (0.48 mg) was added to the reaction solution and stirred for 5 min. Ethyl acetate (0.7 ml) was added with continuous stirring for 5 min followed by the addition of acetonitrile (0.7 ml) and stirred for an additional 5 min. Compound **5** (0.1 g) was added to the mixture and stirred continuously for 3 h until it was completely dissolved and the solution was assessed by thin layer chromatography. An aqueous saturated mixture (10 ml) of hydrogen carbonate and sodium thiosulfate (1:1 v/v) was poured into the solution to quench the reaction. The reaction solution was extracted with ethyl acetate and dried with anhydrous sodium sulfate. The solvent was removed at reduced pressure and the mixture was subjected to silica gel column chromatography using a hexane-ethyl acetate mixture as an elution solvent to obtain **6** (yield 58%). [1]H NMR (in Acetone-d6) $\delta$: 2.24 (3H, s), 3.82 (3H, s), 3.83 (3H, s), 3.95 (3H, s), 4.60 (1H, d, J = 7.6 Hz), 6.22 (1H, d, J = 7.6 Hz), 7.06 (1H, dd, J = 8.1 Hz, 1.8 Hz), 7.09 (1H, d, J = 8.1 Hz), 7.25 (1H, d, J = 1.8 Hz), 7.50 (1H, s), 7.61 (1H, s), 9.88 (1H, s); [13]C-NMR $\delta$: 20.3, 53.0, 55.3, 56.1, 56.3, 88.0, 111.2, 113.6, 118.9, 121.4, 123.8, 126.9, 132.7, 139.1, 141.0, 145.9, 152.4, 153.8, 168.8, 171.1, 190.7.

Compound **6** (0.1 mg) was mixed with pyridine (5 ml), [2-[13]C] malonic acid (38.4 mg) and a small amount of piperidine (six drops) and incubated at 60°C for 2 h. The reaction mixture was acidified with hydrochloric acid and then extracted with ethyl acetate. The organic layer was washed with water and brine and dried with anhydrous sodium sulfate. Removal of the solvent at reduced pressure yielded a crude form of **7**.

At 0°C and under argon gas, dry methanol (10 ml) was mixed with trimethylsilyl chloride (0.1 ml) and the solution was stirred for 20 min. Crude **7** (0.1 g) was added to the solution and heated at reflux (90°C) for 1 h. After cooling to room temperature, the reaction mixture was dissolved in dichloromethane and washed with distilled water and brine prior to being dried using anhydrous sodium sulfate. The organic solvent was evaporated under reduced pressure and then it was purified by silica gel column chromatography using hexane-ethyl acetate mixture as an elution solvent to obtain **8** (yield 19%). [1]H NMR (in Acetone-d6) $\delta$: 3.78 (3H, s), 3.81 (3H, s), 3.84 (3H, s), 3.92 (3H, s), 4.47 (1H, d, J = 7.9 Hz), 6.04 (1H, d, J = 7.9 Hz), 6.66 (1H, dd, J = 15.9 Hz, 161.9 Hz), 6.85 (1H, d, J = 8.1 Hz), 6.91 (1H, dd, J = 8.1 Hz, 1.8 Hz), 7.10 (1H, d, J = 1.9 Hz), 7.29 (1H, s), 7.33 (1H, s), 7.63 (1H, dd, J = 16.0 Hz, 2.9 Hz); [13]C-NMR $\delta$: 51.6, 53.0, 56.0, 56.3, 56.5, 88.4, 110.8, 113.4, 116.3, 116.3, 117.2, 120.2, 129.4, 132.0, 145.1, 145.8, 148.0, 148.6, 151.0, 167.4, 171.7.

Lithium aluminium hydride (28 mg) and anhydrous tetrahydrofuran (5 ml) were mixed together in a round-bottom flask under nitrogen gas. Anhydrous tetrahydrofuran solution (5 ml) was added dropwise

to **8** (71.5 mg) while stirring and mixed for 1 h. The reaction solution was cooled to 0°C and quenched by slowly adding a mixture of methanol and tetrahydrofuran (1:5 v/v) (1.2 ml). The reaction mixture was poured on dry ice (approx. 1 g). After the addition of water, the reaction mixture was extracted with ethyl acetate and washed with brine. The organic solvent was dried with anhydrous sodium sulfate and the solvent was removed under reduced pressure. The reaction mixture was purified by silica gel column chromatography using a hexane-ethyl acetate mixture as an elution solvent to obtain [13]C-labelled β-5 dilignol [13]**C-II** (yield 41%). [1]H NMR (in Acetone-d6) δ: 3.54 (1H, *q*, 6.3 Hz), 3.71 (1H, *m*), 3.82 (3H, *s*), 3.86 (3H, *s*), 4.20 (1H, *m*), 5.57 (1H, *d*, *J* = 6.6 Hz), 6.24 (1H, *ddt*, *J* = 5.6 Hz, 15.6 Hz, 150.4 Hz), 6.54 (1H, *d*, *J* = 4.8 Hz), 6.81 (1H, *d*, *J* = 8.1 Hz), 6.89 (1H, *dd*, *J* = 8.3 Hz, 1.8 Hz), 6.95 (1H, *s*), 6.98 (1H, *s*), 7.04 (1H, *d*, *J* = 1.8 Hz); [13]C-NMR δ: 54.8, 56.3, 60.5, 64.6, 88.5, 110.5, 111.7, 115.7, 116.1, 119.6, 128.4, 130.2, 131.5, 134.4, 145.2, 147.3, 148.4, 149.0.

**II** was also obtained in the same manner starting with **4** (yield 38%). δ: 3.54 (1H, *q*, 6.3 Hz), 3.71 (1H, *m*), 3.82 (3H, *s*), 3.86 (3H, *s*), 4.20 (1H, *m*), 5.57 (1H, *d*, *J* = 6.6 Hz), 6.24 (1H, *dt*, *J* = 5.2 Hz, 15.6 Hz), 6.53 (1H, *d*, *J* = 16.0 Hz), 6.81 (1H, *d*, *J* = 8.1 Hz), 6.89 (1H, *dd*, 8.3 Hz, 1.8 Hz), 6.95 (1H, *s*), 6.98 (1H, *s*), 7.04 (1H, *d*, 1.8 Hz); [13]C-NMR δ: 54.7, 56.2, 56.3, 63.4, 64.6, 88.5, 110.4, 111.6, 115.6, 116.0, 119.5, 128.3, 130.4, 130.5, 131.9, 134.3, 145.1, 147.2, 148.3, 149.0.

ESI-TOF-MS (Mariner 2, Applied Biosystems) $m/z$ 381.12806 [**II** + Na]$^+$, calcd. for $C_{20}H_{22}O_6Na$, 381.13086.

ESI-TOF-MS $m/z$ 382.13205 [[13]**C-II** + Na] $^+$, calcd. for [13]C $C_{19}H_{22}O_6Na$, 382.13421.

## 2.2. Enzymatic dehydrogenative polymerization

An acetone solution containing [13]**C-II** (20 mg) was mixed with water (20 ml) and an aqueous horseradish peroxidase solution (0.01 mg ml$^{-1}$, 4 ml, 15.6 unit) and stirred. To start the dehydrogenative polymerization, a hydrogen peroxide solution (0.1%, 1 ml) was added to the mixture. After 3 h, the horseradish peroxidase (4 ml) and hydrogen peroxide were added and incubated for 24 h. A solution of distilled water and catalase (0.01 mg ml$^{-1}$, 8 ml) was added to quench the reaction. The reaction mixture was then freeze-dried to obtain the [13]C-labelled enzymatic dehydrogenative polymer (**DHP-[13]C-II**).

The enzymatic dehydrogenative polymer (**DHP-II**) using **II** was prepared in the same manner.

## 2.3. NMR

**DHP-II** and **DHP-[13]C-II** were dissolved in 0.5 ml of CD$_3$OD (2.4%). The [13]C NMR spectra were recorded on a Bruker Avance 600 ([1]H 600 MHz, [13]C 150 MHz) spectrometer equipped with a cryoprobe. The central methanol solvent peak was used as an internal reference ($\delta_C$ 49.0, $\delta_H$ 3.31 ppm). The standard Bruker implementation for HSQC experiments was used. Acetylated **DHP-[13]C-II** was dissolved in 0.5 ml CDCl$_3$ and the central chloroform solvent peak was used as an internal reference ($\delta_C$ 77.0, $\delta_H$ 7.26 ppm).

# 3. Results and discussion

**DHP-II** was dissolved in deuterium labelled methanol for NMR analysis. The two-dimensional NMR (HSQC) spectrum of **DHP-II** is shown in figure 3. The signals for $C_\beta$–$H_\beta$, $C_\gamma$–$H_\gamma$, $C_\gamma$–$H_{\gamma'}$ and $C_\alpha$–$H_\alpha$ correlations in β-5′ structures were observed at $\delta_C/\delta_H$ of 55.1–55.2/3.48–3.56, 63.9–64.8/3.63–4.16, 63.9–64.8/3.63–4.16, and 89.4/5.30–5.68, respectively [12–17]. Surprisingly, the signals for $C_\gamma$–$H_\gamma$ of the β′-β′ structure also appeared at $\delta_C/\delta_H$ of 71.9/3.40–3.75 and 71.9/4.12–4.20. This finding implied that a β′ radical was generated due to the cleavage of the α-$O$-4′ linkage in **II** and that the two radicals performed a coupling reaction to create the β′-β′ structure.

To confirm the hypothesis, **II** labelled with [13]C at the β′-position ([13]**C-II**) was synthesized and subjected to enzymatic dehydrogenative polymerization in the same manner as described above (figure 3). If the β′ carbon participated in the polymerization, then new signals derived from the [13]C labelled β′ carbon would be detected. [13]**C-II** was created in an eight-step reaction starting with vanillin. By using [2-[13]C] malonic acid, labelling at the β′-position was achieved (figure 2).

The HSQC spectrum of the [13]C labelled enzymatic dehydrogenative polymer (**DHP-[13]C-II**) is shown in figure 2b. As expected, new signals were detected when **DHP-[13]C-II** was compared with the **DHP-II** reference molecule. At $\delta_C/\delta_H$ of 55.1–55.3/2.89–2.93, 84.2–87.3/4.00–4.38 and 89.3–89.4/3.72–3.74, the signals corresponding to $C_{\beta'}$-$H_{\beta'}$ of β′-β′, β′-$O$-4 and dibenzodioxin structures were observed [12–19].

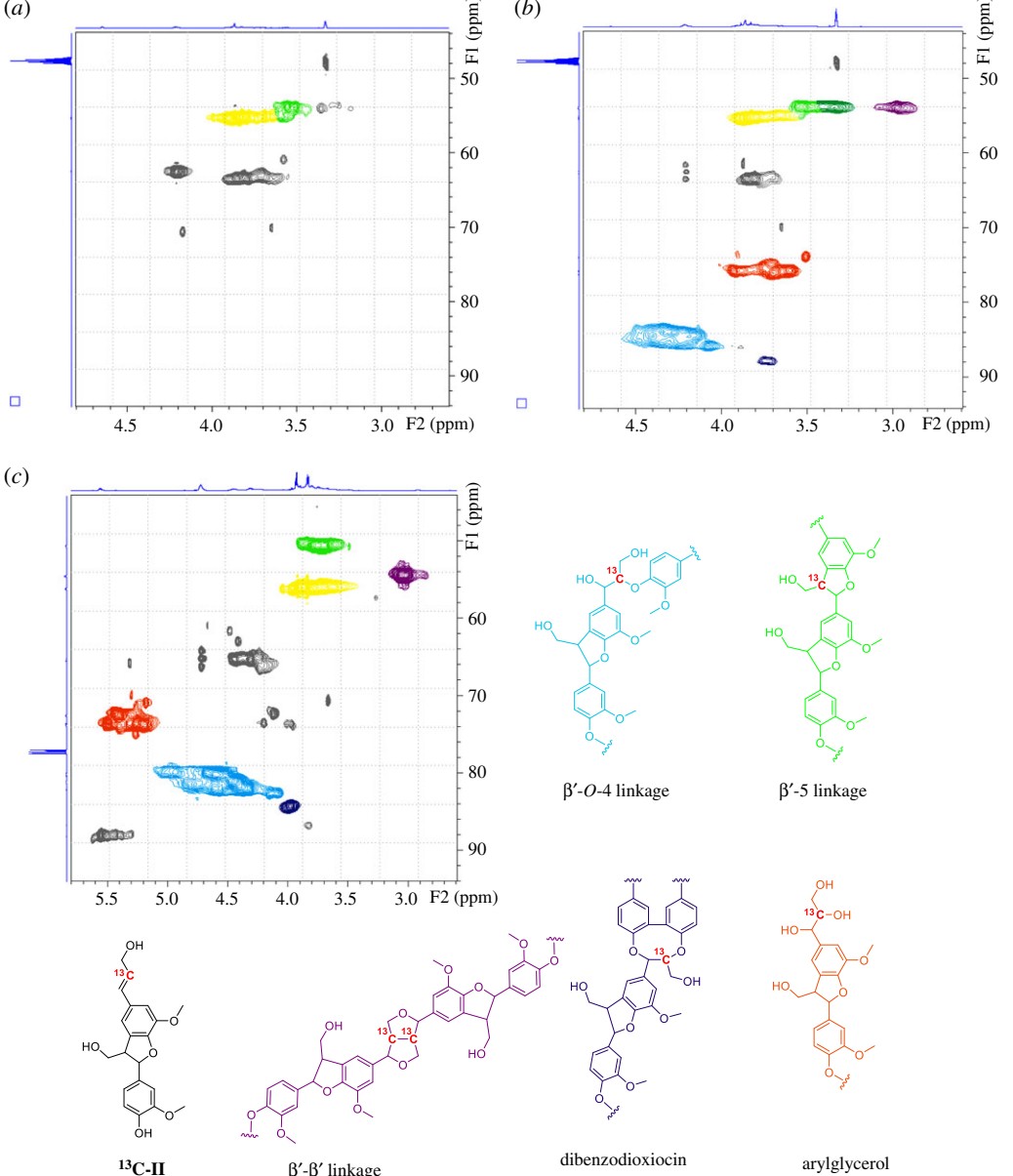

**Figure 3.** HSQC NMR spectra of enzymatic dehydrogenative polymer. (*a*) Prepared from **II**, (*b*) prepared from **[13]C-II**, (*c*) acetylated enzymatic dehydrogenative polymer prepared from **[13]C-II**.

In addition, an increase in the signal at 55.1–55.3/3.44–3.56 suggested that β′-5′ structures were generated during the reaction. This result suggests that cleavage of the α-*O*-4′ linkage of **[13]C-II** should occur during the enzymatic dehydrogenative polymerization.

Currently, the cleavage mechanism of α-*O*-4′ linkage is unclear. There are two possibilities, which involved homolytic and heterolytic cleavages (figure 1*b*), although it is unknown which route is predominant.

There is a possibility that the double bond between $C_\alpha$ and $C_\beta$ of **II** is the result of an enzymatic reaction, yielding phenylcoumaron. With the generation of the double bond, two aromatic rings are connected through the π bond. In the case of phenylcoumaron, however, the radical generated at the phenolic oxygen (C4-O) does not transfer to the β′-position through hybrid resonance. Thus, the cleavage route through phenylcoumaron is not considered at present.

In the HSQC spectrum of **DHP-[13]C-II**, large indeterminate signals were detected at $\delta_C/\delta_H$ of 75.2–77.6/3.60–3.93. These signals, which seem to be minor structures, are not normally observed in lignin. However, the signals shifted 69.9–73.8/5.21–5.54 after acetylation (figure 3*c*), therefore, the signals should be derived from the introduced hydroxy group at the β′-position leading to the guaiacylglycerol unit. The presence of the arylglycerol structure in lignin has been suggested in a previous study of mild hydrolysis of lignin [20].

Higuchi *et al.* [21] also found the structure in DHP of monolignols. Kilpeläinen *et al.* [22] investigated minor structural units of acetylated hardwood and softwood lignin via two-dimensional NMR spectroscopy and identified the guaiacylglycerol unit by correlation at $\delta_C/\delta_H$ of 72.7/5.41. This correlation is consistent with the findings herein. The reaction mechanism behind the formation of the guaiacylglycerol structure is still unclear; however, it may be due to a coupling reaction between the β′radical of $^{13}$**C-II** and a hydroxy radical originating from hydrogen peroxide.

In the acetylated HSQC spectrum of **DHP-$^{13}$C-II**, the signals corresponding to $C_{β'}$-$H_{β'}$ of β′-β′ ($\delta_C/\delta_H$ of 54.5/2.97–3.04), β′-*O*-4 ($\delta_C/\delta_H$ of 79.2–82.0/4.09–4.49) and dibenzodioxocin ($\delta_C/\delta_H$ of 84.4/3.95–3.96) structures also appeared [23–26] (figure 3*c*).

It has long been accepted that only phenolic oxygen or the ring 5 position of β-5′ dilignol (**II**) can react with and grow the lignin molecule. However, in this study, we demonstrated that the double bond of **II** is also involved in the reaction. This is the first report to propose this reaction mechanism.

In our previous study, we investigated the behaviour of the dilignols during enzymatic dehydrogenative polymerization and proposed that a radical transfer between dilignols occurs during lignification [7,8]. In the radical transfer system, the radicalized **II** donates a radical to **I** and **III**, which suggests that **II** has a specific reactivity and plays an important role in the lignin growing process. The reaction mechanism of growing lignols and the resultant lignin structure are still ambiguous. To understand the lignification process and lignin structure, the reactivity of mono-, di- and oligo-lignols needs to be further elucidated.

Lignin has a highly complicated structure; therefore, we believe that elucidating its structure based on the experimental results of lignin biosynthesis will lead to its effective use.

Under mild conditions using an enzyme, the α-*O*-4′ linkage of the β-5′ structure was cleaved to generate a new phenolic reaction site. If this reaction site can be used to introduce new functional groups industrially, a novel functionalized lignin can be obtained.

Data accessibility. All data generated or analysed during this study are included in this published article.

Authors' contributions. Y.M. mainly planned this research and also analysed all data obtained in this research. Y.M. was a major contributor in writing the manuscript and corresponding author. Y.O. performed most experiments and analysed all data obtained in this research. D.A. and K.F. also analysed the NMR data. All authors read and approved the final manuscript.

Competing interests. The authors declare that they have no competing interests

Funding. This work was supported by the Japan Society for the Promotion of Science KAKENHI (grant no. 17H03842).

Acknowledgements. The authors wish to thank KentaWatanabe for supporting the synthesis of the dilignols.

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
