## [Reviewer comments · Royal Society Open Science]

Review History

RSOS-190445.R0 (Original submission)

Review form: Reviewer 1

Is the manuscript scientifically sound in its present form?

No

Are the interpretations and conclusions justified by the results?

Yes

Is the language acceptable?

Yes

Is it clear how to access all supporting data?

Yes

Do you have any ethical concerns with this paper?

No

Have you any concerns about statistical analyses in this paper?

No

Recommendation?

Major revision is needed (please make suggestions in comments)

Comments to the Author(s)

In this manuscript, the reaction between two monolignols by enzymatically oxidation is analyzed into dilignol is analyzed. one of the final production is found to show the reactivity for new radical reaction. It is an interesting founding, but as for current version, it is more suitable for a technical report as intermediate results instead of full published paper. This manuscript lack the story, like the importance of this founding is unclear to the reader, the potential applications, the conclusion is not clear enough, the confirming reaction from this finding is missing. It is not suggested to be published in this current version.

Review form: Reviewer 2

Is the manuscript scientifically sound in its present form?

Yes

Are the interpretations and conclusions justified by the results?

Yes

Is the language acceptable?

Yes

Is it clear how to access all supporting data?

Yes

Do you have any ethical concerns with this paper?

No

Have you any concerns about statistical analyses in this paper?

I do not feel qualified to assess the statistics

Recommendation?

Accept as is

Comments to the Author(s)

The authors have proposed a novel mechanism in lignin polymerisation. The biosynthesis of lignin has attracted the attention of researchers due to its complex nature. It is believed to proceed by repeated radical coupling reaction of lignols by enzymatic dehydrogenative polymerisation mechanism. In the present paper the authors have proposed cleavage of a covalent linkage and creation of new radical coupling site. They have synthesised a labelled intermediate to support the proposed mechanism and addition of lignol radicals in dilignol formation. The paper is well written and free from typographical errors. [

Decision letter (RSOS-190445.R0)

24-Apr-2019

Dear Dr Matsushita:

Title: Unexpected polymerization mechanism of dilignol in the lignin growing
Manuscript ID: RSOS-190445

The editor assigned to your manuscript has now received comments from reviewers. We would like you to revise your paper in accordance with the referee and Subject Editor suggestions which can be found below (not including confidential reports to the Editor). Please note this decision does not guarantee eventual acceptance.

Please submit your revised paper before 17-May-2019. Please note that the revision deadline will expire at 00.00am on this date. If we do not hear from you within this time then it will be assumed that the paper has been withdrawn. In exceptional circumstances, extensions may be possible if agreed with the Editorial Office in advance. We do not allow multiple rounds of revision so we urge you to make every effort to fully address all of the comments at this stage. If deemed necessary by the Editors, your manuscript will be sent back to one or more of the original reviewers for assessment. If the original reviewers are not available we may invite new reviewers.

Please also include the following statements alongside the other end statements. As we cannot publish your manuscript without these end statements included, if you feel that a given heading is not relevant to your paper, please nevertheless include the heading and explicitly state that it is not relevant to your work.

- Ethics statement

Please clarify whether you received ethical approval from a local ethics committee to carry out your study. If so please include details of this, including the name of the committee that gave consent in a Research Ethics section after your main text. Please also clarify whether you received informed consent for the participants to participate in the study and state this in your Research Ethics section.

OR

Please clarify whether you obtained the necessary licences and approvals from your institutional animal ethics committee before conducting your research. Please provide details of these licences and approvals in an Animal Ethics section after your main text.

OR

Please clarify whether you obtained the appropriate permissions and licences to conduct the fieldwork detailed in your study. Please provide details of these in your methods section.

RSC Associate Editor:

Comments to the Author:

According to the report of Reviewer 1, further work is required for the work to be suitable for full paper format. Further information is needed in the introduction and conclusion sections and potential applications should be explained if possible.

RSC Subject Editor:

Comments to the Author:

(There are no comments.)

Reviewers' Comments to Author:

Reviewer: 1

Comments to the Author(s)

In this manuscript, the reaction between two monolignols by enzymatically oxidation is analyzed into dilignol is analyzed. one of the final production is found to show the reactivity for new radical reaction. It is an interesting founding, but as for current version, it is more suitable for a technical report as intermediate results instead of full published paper. This manuscript lack the story, like the importance of this founding is unclear to the reader, the potential applications, the conclusion is not clear enough, the confirming reaction from this finding is missing. It is not suggested to be published in this current version.

Reviewer: 2

Comments to the Author(s)

The authors have proposed a novel mechanism in lignin polymerisation. The biosynthesis of lignin has attracted the attention of researchers due to its complex nature. It is believed to proceed by repeated radical coupling reaction of lignols by enzymatic dehydrogenative polymerisation mechanism. In the present paper the authors have proposed cleavage of a

covalent linkage and creation of new radical coupling site. They have synthesised a labelled intermediate to support the proposed mechanism and addition of lignol radicals in dilignol formation. The paper is well written and free from typographical errors. [

Author's Response to Decision Letter for (RSOS-190445.R0)

See Appendix A.

RSOS-190445.R1 (Revision)

Review form: Reviewer 2

Is the manuscript scientifically sound in its present form?

Yes

Are the interpretations and conclusions justified by the results?

Yes

Is the language acceptable?

Yes

Is it clear how to access all supporting data?

Not Applicable

Do you have any ethical concerns with this paper?

No

Have you any concerns about statistical analyses in this paper?

I do not feel qualified to assess the statistics

Recommendation?

Accept with minor revision (please list in comments)

Comments to the Author(s)

The authors have revised the manuscript by introducing few opening remarks in the introduction section and in conclusion. In order to highlight the reaction between two monolignols by enzymatic oxidation to dilignols, the revised statement starts with 'Because' which needs to be changed. Similar minor mistakes are still in the manuscript and need careful editing.

Other sections in the manuscript are same as in the original manuscript.

The manuscript needs minor revision .

Decision letter (RSOS-190445.R1)

12-Jun-2019

Dear Dr Matsushita:

Title: Unexpected polymerization mechanism of dilignol in the lignin growing
Manuscript ID: RSOS-190445.R1

Thank you for submitting the above manuscript to Royal Society Open Science. On behalf of the Editors and the Royal Society of Chemistry, I am pleased to inform you that your manuscript will be accepted for publication in Royal Society Open Science subject to minor revision in accordance with the referee suggestions. Please find the reviewers' comments at the end of this email.

The reviewers and handling editors have recommended publication, but also suggest some minor revisions to your manuscript. Therefore, I invite you to respond to the comments and revise your manuscript. I apologise that this has taken longer than usual.

Because the schedule for publication is very tight, it is a condition of publication that you submit the revised version of your manuscript before 21-Jun-2019. Please note that the revision deadline will expire at 00.00am on this date. If you do not think you will be able to meet this date please let me know immediately.

Supplementary files will be published alongside the paper on the journal website and posted on

the online figshare repository (<https://figshare.com>). The heading and legend provided for each supplementary file during the submission process will be used to create the figshare page, so please ensure these are accurate and informative so that your files can be found in searches. Files on figshare will be made available approximately one week before the accompanying article so that the supplementary material can be attributed a unique DOI.

Best wishes,

Dr Laura Smith
Publishing Editor, Journals

RSC Associate Editor:
Comments to the Author:
(There are no comments.)

RSC Subject Editor:
Comments to the Author:
(There are no comments.)

Reviewer comments to Author:
Reviewer: 2

Comments to the Author(s)

The authors have revised the manuscript by introducing few opening remarks in the introduction section and in conclusion. In order to highlight the reaction between two monolignols by enzymatic oxidation to dilignols, the revised statement starts with 'Because' which needs to be changed. Similar minor mistakes are still in the manuscript and need careful editing.

Other sections in the manuscript are same as in the original manuscript.
The manuscript needs minor revision .

Author's Response to Decision Letter for (RSOS-190445.R1)

See Appendix B.

Decision letter (RSOS-190445.R2)

01-Jul-2019

Dear Dr Matsushita:

Title: Unexpected polymerization mechanism of dilignol in the lignin growing
Manuscript ID: RSOS-190445.R2

It is a pleasure to accept your manuscript in its current form for publication in Royal Society Open Science. The chemistry content of Royal Society Open Science is published in collaboration with the Royal Society of Chemistry.

RSC Associate Editor

Comments to the Author:

After considering the revisions, we are happy to accept this revised manuscript for publication. In the title, please change "Unexpected" to "Unexpeced".

Reviewer(s)' Comments to Author:

Appendix A

Dear Dr. Ellis Wilde
Royal Society Open Science

Thank you for your e-mail. The response is described as follows.

Reviewer: 1

[Suggestion and Comments]

In this manuscript, the reaction between two monolignols by enzymatically oxidation is analyzed into dilignol is analyzed. one of the final production is found to show the reactivity for new radical reaction. It is an interesting founding, but as for current version, it is more suitable for a technical report as intermediate results instead of full published paper. This manuscript lack the story, like the importance of this founding is unclear to the reader, the potential applications, the conclusion is not clear enough, the confirming reaction from this finding is missing. It is not suggested to be published in this current version.

[Answer]

Thank you for your reviewing our manuscript. We revised our paper basically according to your suggestion and comments.

We added the story of this study to state the importance and application of this founding for readers in introduction and conclusion as follows.

Introduction:

Because lignocellulosic biomass has a carbon neutral characteristic, it must be actively utilized for the future sustainable society. Among the main components of lignocellulosic biomass, polysaccharide components such as cellulose are used for fibers and paper, but the use of lignin is hardly advanced. One of the reasons is the complexity of the chemical structure. It is well known that lignin is a phenolic natural polymer and does not contain a definite repeating unit [1,2]. In order to make effective use of lignin, it is first necessary to understand its structure. In this study, we approached the chemical structure of lignin from the viewpoint of biosynthesis.

Conclusion:

Finally, this study expressed a novel aspect of the structural variety of lignin. The α -O-4' linkage of β -5' structure can be cleaved under the mild condition to generate phenolic oxygen enabling further reactions. This finding will help us not only to

understand the biosynthetic process of lignin but also to do further utilization of lignin via biological and chemical techniques.

Reviewer: 2

[Suggestion and Comments]

The authors have proposed a novel mechanism in lignin polymerisation. The biosynthesis of lignin has attracted the attention of researchers due to its complex nature. It is believed to proceed by repeated radical coupling reaction of lignols by enzymatic dehydrogenative polymerisation mechanism. In the present paper the authors have proposed cleavage of a covalent linkage and creation of new radical coupling site. They have synthesised a labelled intermediate to support the proposed mechanism and addition of lignol radicals in dilignol formation. The paper is well written and free from typographical errors.

[Answer]

I thought the response were not required.

We would be grateful if the manuscript could be re-reviewed and considered for publication in *Royal Society Open Sciences*.

Yours sincerely,

Yasuyuki Matsushita
Graduate School of Bioagricultural
Sciences
Nagoya University
E-mail: ysmatsu@agr.nagoya-u.ac.jp

Appendix B

Furo-cho, Chikusa-ku, Nagoya
464-8601, Japan
17. June, 2019

Dear Editor of Royal Society Open Sciences

Thank you for reviewing our manuscript. We have revised the manuscript according to the reviewer's suggestions. Please find enclosed the revised version entitled "Unexpected polymerization mechanism of dilignol in the lignin growing" which was assigned manuscript ID RSOS-190445R1.

We would be grateful if the manuscript could be re-reviewed and considered for publication in *Royal Society Open Sciences*.

Yours sincerely,

Yasuyuki Matsushita
Graduate School of Bioagricultural Sciences
Nagoya University
E-mail: ysmatsu@agr.nagoya-u.ac.jp

Answer to Reviewer 2

Thank you for your reviewing our manuscript. We revised our paper basically according to your suggestion and comments.

Comments to the Author(s)

The authors have revised the manuscript by introducing few opening remarks in the introduction section and in conclusion. In order to highlight the reaction between two monolignols by enzymatic oxidation to dilignols, the revised statement starts with 'Because' which needs to be changed. Similar minor mistakes are still in the manuscript and need careful editing. Other sections in the manuscript are same as in the original manuscript.

The manuscript needs minor revision.

[Answer]

Thank you for your comments. We rewrote the Introduction and Conclusion sections as follows, Introduction:

Lignocellulosic biomass exhibits a carbon neutral characteristic and thus should be utilized effectively to reduce carbon dioxide emission, which is one of the causes of global warming. Lignocellulosic biomass is composed of polysaccharides, such as cellulose, and a phenolic polymer, lignin. While cellulose is used in the production of fibres and paper, the use of lignin is limited. One of the reasons is the complexity of its chemical structure. It is well known that lignin is a phenolic natural polymer and does not contain a definite repeating unit [1,2]. Therefore, understanding the structure of lignin is essential for its effective use. In this study, we carry out lignin biosynthesis to help understand its structure.

Conclusion:

Lignin has a highly complicated structure; therefore, we believe that elucidating its structure based on the experimental results of lignin biosynthesis will lead to its effective use.

Under mild conditions using an enzyme, the α -O-4' linkage of the β -5' structure was cleaved to generate a new phenolic reaction site. If this reaction site can be used to introduce new functional groups industrially, a novel functionalized lignin can be obtained.

We would be grateful if the manuscript could be re-reviewed and considered for publication in *Royal Society Open Sciences*.

Yours sincerely,

Yasuyuki Matsushita
Graduate School of Bioagricultural Sciences
Nagoya University
E-mail: ysmatsu@agr.nagoya-u.ac.jp